# Effect of Traditional Chinese Exercises on Patients with Chronic Heart Failure (TCE-HF): A Systematic Review and Meta-Analysis

**DOI:** 10.3390/jcm12062150

**Published:** 2023-03-09

**Authors:** Qinyi Bao, Shuxin Lei, Shitian Guo, Zhuo Wang, Jiaye Yu, Yayu You, Ning Zhang, Xiaojie Xie

**Affiliations:** 1Department of Cardiology, Second Affiliated Hospital, Zhejiang University School of Medicine, Hangzhou 310009, China; 2Department of Cardiology, First People’s Hospital, Zhejiang University School of Medicine, Hangzhou 310006, China

**Keywords:** heart failure, cardiac rehabilitation, traditional Chinese exercise, exercise capacity, cardiac function

## Abstract

Exercise-based cardiac rehabilitation is safe and effective for chronic heart failure (CHF) patients. The present study aimed to investigate the effects of traditional Chinese exercise (TCE) on patients with CHF and the impact of exercise types and duration. Evaluation of randomized controlled trials (RCTs) of TCE in patients with CHF published since 1997 from PubMed, Embase, Web of Science, the Cochrane Library, Chongqing VIP, Wanfang Databases, and the China National Knowledge was performed. A total of 41 RCTs, including 3209 patients with CHF, were included. It showed that TCE significantly increased 6-min walk distance (6MWD) [mean difference (MD) = 72.82 m, *p* < 0.001] and left ventricular ejection fraction (MD = 5.09%, *p* < 0.001), whereas reduced B-type natriuretic peptide (BNP) (MD = −56.80 pg/mL, *p* < 0.001), N-terminal pro-BNP (MD = −174.94 pg/mL, *p* < 0.05), and Minnesota Living with Heart Failure Questionnaire scores (MD = −11.31, *p* < 0.001). However, no significant difference was found in the effects of TCE on peak oxygen consumption. The increase in TCE weekly duration and program duration significantly improved 6MWD (MD = 71.91 m, *p* < 0.001; MD = 74.11 m, *p* < 0.001). The combination of TCE and conventional aerobic exercise significantly improved 6MWD (MD = 19.86 m, *p* < 0.005). TCE improves exercise capacity, cardiac function, and quality of life in patients with CHF, which might be an optimal and available pattern of exercise-based cardiac rehabilitation.

## 1. Introduction

Heart failure (HF), the terminal stage of various cardiovascular diseases, is a complex, serious clinical syndrome. Patients with chronic heart failure (CHF) usually suffer from lower exercise tolerance, reduced quality of life (QoL), mental health, and a high risk of hospitalizations [1,2]. The 1-year mortality and 1-year hospitalization rate of patients with CHF are reported to be 7.2% and 31.9%, respectively [3]. The mortality may be even higher in patients with heart failure with reduced ejection fraction (HFrEF) [4]. There is increasing awareness that exercise-based cardiac rehabilitation (ExCR) is safe and effective for patients with CHF. Clinical trials and meta-analyses have also shown that ExCR could improve exercise tolerance, reduce all-cause and HF-related hospitalization rates, and enhance QoL [2,5,6,7].

The effect of ExCR may rely on high compliance with the exercise program [8]. Therefore, traditional Chinese exercise (TCE) could be an alternative option. TCE is a type of low-intensity aerobic exercise, which mainly includes Tai Chi, Baduanjin (eight silken movements), Yijinjing (changing tendons exercises), Wuqinxi (five animals play), and Liuzijue (six tips). The characteristics of each type of TCE are detailed in Appendix A TCE can be practiced with minimal time investment without the limitation of place, time, or specialized equipment. This characteristic makes TCE to be easily incorporated into daily routines and eventually improves patient adherence. With the increasing global popularity of TCE, it is more widely utilized in cardiac rehabilitation (CR) programs for HF patients. Reported benefits of TCE include improvements in clinical symptoms, QoL, exercise tolerance, cardiac function, sleep quality, and the alleviation of depressive symptoms in patients with HF [9,10,11,12,13,14,15]. Several randomized controlled trials (RCTs) reported that left ventricular ejection fraction (LVEF) and N-terminal pro-B natriuretic peptide (NT-proBNP) had no noticeable improvement within 3 months of Liuzijue exercise [16]. However, B-type natriuretic peptide (BNP) levels could dramatically decline after 1 year of exercise [17,18]. Additionally, there was an RCT to evaluate the therapeutic effect of Baduanjin which enrolled 63 patients with CHF due to coronary heart disease and ultimately reported no significant difference after 3 months of Baduanjin on plasma NT-proBNP compared to the control group [19].

Considering that consensus regarding the effect of TCE on patients with CHF has not been reached, the primary aim of this systematic review and meta-analysis is to assess the potential benefits of TCE on patients with CHF. The present study further explored whether different types of TCE and various study characteristics had different effects on CR in HF patients, thereby providing references for further application of TCE in HF patients.

## 2. Materials and Methods

This study adheres to the guidelines for systematic reviews according to the Cochrane Handbook and Preferred Reporting Items for Systematic Reviews and Meta-Analyses (PRISMA) [20]. The study protocol has been published previously in INPLASY (registration number: 202190027; DOI number: 10.37766/inplasy2021.9.0027) [21]. There were a few minor changes to the title and the anticipated dates of the literature completion.

### 2.1. Search Strategy

Three reviewers independently conducted a systematic literature search of articles published from the inceptions of the following electronic data sources to 11 August 2022: PubMed, Embase, Web of Science, the Cochrane Library, Chongqing VIP, Wanfang Databases, and the China National Knowledge Infrastructure (CNKI). The search was limited to RCTs with human subjects, which were published in either English or Chinese. Additional clinical trials which were not identified by the electronic search were retrieved by searching the reference lists of relevant articles. A complete search strategy is provided in Appendix A.

### 2.2. Eligibility Criteria

The titles and abstracts of all identified articles were scanned first. Full-text articles using RCTs design, incorporating TCE intervention, among participants with all types of CHF were included. There were no limits according to age, sex, country, outcome, the type of control groups, or TCE intervention type, time, frequency, or intensity. Studies that lacked data for outcome evaluation, had a high bias or were published as reviews, comments, letters, editorials, historical articles, meta-analyses, and guidelines were excluded.

### 2.3. Data Extraction and Management

The following data were independently extracted from the included studies by two reviewers and checked by a third: (1) study characteristics: authors, publication year; (2) participant characteristics: number, age, country; New York Heart Association (NYHA) class; baseline LVEF; (3) description of the intervention and control groups; (4) duration; (5) main outcomes: 6-min walking distance (6MWD), BNP, NT-proBNP, QoL [e.g., Minnesota Living with Heart Failure Questionnaire (MLHFQ), 36-Item Short Form (SF-36)], peak oxygen consumption (peakVO2), LVEF.

### 2.4. Quality Assessment

The risk of bias in the included studies was independently evaluated by two reviewers with the Cochrane Collaboration’s recommended tool [22]. The following domains were assessed: random sequence generation (selection bias); allocation concealment (selection bias); blinding of participants and personnel (performance bias); blinding of outcome assessment (detection bias); incomplete outcome data (attrition bias); selective reporting (reporting bias); and other sources of bias.

### 2.5. Statistical Analysis

The Review Manager (RevMan) software Version 5.3. (Cochrane Collaboration, Copenhagen, Denmark) was used to conduct the meta-analysis. Given that all variables in the included studies were continuous data, the mean difference (MD) or standardized mean difference (SMD) with corresponding 95% confidence intervals (CI) were calculated in this study. The MD was used when all studies assessed the same outcome using the same scale. The sample mean and standard deviation were estimated if the continuous data were reported as median and interquartile range. Specific estimation methods were reported in a previous study [23]. According to the Cochrane Handbook, studies with change-from-baseline outcomes and final measurement outcomes were combined in analysis using the (unstandardized) mean difference method in RevMan [22].

The chi-square test and I^2^ statistic were employed to analyze heterogeneity, with *p* < 0.10 indicating evidence of heterogeneity. The I^2^ statistic was used to measure the degree of heterogeneity, with suggested thresholds for low (25–49%), moderate (50–74%), and high (75%) values. A fixed-effect model was used for low heterogeneity (I^2^ < 50%), and a random-effect model was used for higher heterogeneity. Funnel plots and Begg’s and Egger’s tests were used to detect potential publication bias. Sensitivity analysis was conducted by removing each study individually to estimate the consistency and quality of the results. Subgroup analyses were undertaken to identify potential sources of heterogeneity and investigate the influence of various study characteristics on the observed effect. *p* < 0.05 was statistically significant.

## 3. Results

### 3.1. Search Results

The literature search yielded a total of 424 titles. Following the review of titles and abstracts, 168 publications remained for full-text evaluation. In a more detailed review, 127 additional studies were excluded due to the following reasons: non-RCTs, non-HF, protocol articles, review articles, or other interventions. Finally, 41 studies that fulfilled all inclusion criteria of the TCE-HF study were included in this systematic review and meta-analysis (Figure 1).

### 3.2. Study Characteristics

The 41 RCTs included 3209 patients (1615 for the TCE group and 1594 for the control group). The TCE groups received usual care in addition to different types of TCE, which included Tai Chi (*n* = 19), Baduanjin (*n* = 14), Liuzijue (*n* = 2), and a combination of two of the exercise types (*n* = 6). Usual care (*n* = 36) included routine treatment and nursing, therapeutic drugs, dietary and exercise advice, standard medical supervision, lifestyle guidance, mental health advice, and an education program. A total of 33 studies (80.49%) were conducted in Asian countries [12,13,15,16,17,18,19,24,25,26,27,28,29,30,31,32,33,34,35,36], five (19.23%) in the United States [14,37,38,39,40], one (3.85%) in Italy [11], one (3.85%) in the United Kingdom [10] and one in Sweden [41]. The time and frequency of exercise varied among studies. The weekly exercise volume of the TCE group varied widely between 60 and 420 min (Appendix A). Patients in 10 control groups performed aerobic exercise (including walking training and cycling) for 30–60 min, twice to 14 times a week [11,14,26,27,42,43,44,45,46]. Detailed characteristics are displayed in Appendix A.

### 3.3. Risk of Bias and Sensitivity Analysis

Two reviewers independently assessed the quality of the selected studies according to the Cochrane Collaboration’s tool for RCTs. All studies had a high risk of performance bias, 23 had an unclear risk of detection bias, and two had a high risk of selection bias. A summary of the risk of bias is provided in Appendix A. Visual inspection of the funnel plot for each outcome suggested no visible publication bias (Appendix A). This was further confirmed by Egger’s linear regression asymmetry test for each overall outcome (for 6MWD, *p* = 0.671; for LVEF, *p* = 0.838; for MLHFQ, *p* = 0.068; for BNP, *p* = 0.092; for NT-proBNP, *p* = 0.269; for peakVO2, *p* = 0.437). Sensitivity analyses confirmed the robustness of the results (Appendix A).

### 3.4. Overall Outcomes

#### 3.4.1. Exercise Tolerance

A total of 29 studies were used to assess the effect of TCE on 6MWD, including 2270 patients. Significant improvements were found when comparing exercise tolerance by 6MWD. The meta-analyses showed a significant improvement in 6WMD of 72.82 m (random-effects model: 95% CI 53.49, 92.16; *p* < 0.001) (Figure 2A).

#### 3.4.2. Cardiopulmonary Exercise Capacity

Four studies reported data on cardiopulmonary exercise capacity (peak VO2, mL/min/kg). The intervention of the TCE group in three studies was Tai Chi [14,37,39], and one was Baduanjin [31]. However, no significant change in peak VO2 in the TCE group was found compared with the control group (1.28 mL/min/kg, 95% CI: −0.72, 3.27; *p* > 0.05) (Figure 2B).

#### 3.4.3. Cardiac Function

LVEF has been wildly utilized to evaluate cardiac function. A total of 19 studies in HFrEF patients [12,15,16,19,25,27,28,29,30,32,44,46,47,48,49,50,53,54] and two studies in HFpEF patients [14,34] provided results on LVEF, containing 1703 patients. Pooled results showed significantly improved LVEF in the TCE group compared with that of the control group (random-effects model: MD = 5.09%, 95%CI 2.93, 7.24; *p* < 0.001) (Figure 2C).

#### 3.4.4. Serum Biomarkers

An amount of 12 studies assessed BNP as an outcome indicator, including 803 HFrEF patients [17,18,37,39,44,45,48,53,55] and 207 HFpEF patients [14,34]. Four Tai Chi studies [11,41,47,54], four Baduanjin studies [12,19,27,46] and one Liuzijue study [16] assessed NT-proBNP as an outcome indicator in 607 HFrEF patients. The meta-analyses demonstrated that TCE was significantly associated with decreased BNP (fixed-effects model: MD = −56.80 pg/mL, 95%CI −78.26, −35.33; *p* < 0.001) (Figure 3A) and decreased NT-proBNP (random-effects model: MD = −174.94 pg/mL, 95% CI −318.14, −31.73; *p* < 0.05) (Figure 3B).

#### 3.4.5. QoL

An amount of 21 studies reported data on MLHFQ, with lower scores indicating better QoL. Given the absence of the total MLHFQ scores, two studies were excluded from the analysis [30,50]. The aggregate results suggested that TCE was associated with a significantly decreased MLHFQ score (random-effects model: MD = −9.21, 95%CI −10.99, −7.42; *p* < 0.001) (Figure 3C), which indicated that TCE could improve the QoL of patients with CHF.

### 3.5. Subgroup Analysis

#### 3.5.1. TCE Types

Few meta-analyses explored the effect of different types of TCE on patients with CHF; thus, a subgroup analysis was performed to compare the effects of different kinds of TCE on 6MWD with usual care. A total of 10 studies of Tai Chi, three of Baduanjin, and two of Liuzijue were included, with 891, 243, and 75 patients, respectively. The subgroup analyses showed that Tai Chi, Baduanjin, and Liuzijue could significantly lead to an increase of 65.30 m (random-effects model: 95% CI 42.71, 87.90; *p* < 0.005), 63.31 m (random-effects model: 95% CI 36.53, 90.10; *p* < 0.001), and 90.58 m (random-effects model: 95% CI 43.85, 137.31; *p* < 0.001) on 6WMD, respectively (Figure 4).

#### 3.5.2. TCE Duration

The weekly duration varied widely from 60 to 420 min per week, while the total program duration varied from 3 to 12 months. Therefore, a subgroup analysis was performed to assess the effect of TCE duration on CHF compared with usual care. In terms of weekly time, subgroup analyses showed that patients of the TCE group who performed over 150 min weekly had significantly longer 6MWD of 71.91 m (random-effects model: 95% CI 55.26, 88.56; *p* < 0.001) compared to those who exercised between 60 to 150 min weekly (54.77 m, 95% CI 27.40, 82.15; *p* < 0.001) (Figure 5A). TCE with a program duration of ≥6 months significantly elongated 6MWD (random-effects model: 74.11 m; 95% CI 56.13, 92.09; *p* < 0.001) (Figure 5B).

#### 3.5.3. Exercise Program

The preponderance compared the rehabilitation effect between TCE and usual care among the included studies. However, six studies compared TCE with traditional aerobic exercise [14,26,27,36,43,44], and two studies compared combined exercise programs (TCE and conventional aerobic exercise) with aerobic exercise alone [11,46]. Therefore, subgroup analysis was performed to compare the effect of different exercise programs on the exercise capacity of patients with CHF. Patients who performed TCE had longer 6MWD than those who only received usual care (random-effects model: 68.46 m; 95% CI 55.75, 81.17; *p* < 0.001) or aerobic exercise (random-effects model: 95.57 m; 95% CI 34.96, 156.19; *p* < 0.001) (Figure 6). In addition, a combined exercise program was more effective than conventional aerobic exercise alone in improving 6MWD (random-effects model: 19.86 m; 95% CI 7.65, 32.06; *p* < 0.05) (Figure 6).

## 4. Discussion

HF has been recognized as a major global public health issue with high mortality, morbidity, and a heavy medical burden. In the past few decades, the prognosis of HF patients has improved; however, this improvement remains poor and confined [51]. This meta-analysis was conducted to evaluate the benefits of TCE interventions among patients with CHF. This study extended previous systematic reviews and meta-analyses by exploring the potential effect of different types of TCE and exercise duration on HF patients. The meta-analysis indicated that TCE led to significantly better exercise capacity, improved cardiac function, higher QoL, and lower serum BNP/NT-pro BNP levels. Longer exercise duration (>150 min weekly, ≥6 months) could lead to even better improvements in 6WMD. TCE or combined exercise programs (TCE and conventional aerobic exercise) can improve exercise tolerance to a greater extent than aerobic exercise alone.

The results of 6WMD from this study suggested that TCE could improve the exercise tolerance of HF patients. Our findings are consistent with other systematic reviews and meta-analyses of TCE for HF patients [52,57,58]. However, a meta-analysis has reported that Tai Chi did not significantly improve 6MWD in patients with CHF [59]. This inconsistent result might be due to the shorter weekly and program duration, which were less than 150 min and 6 months, respectively. Muscle strength is of good predictive value in the long-term survival of patients with CHF. The reduction of muscle mass and shift of muscle fiber are major muscle abnormalities in CHF. Training has been shown to resolve the alteration in mitochondrial content, increase the cytochrome oxidase activity in patients with CHF, and thereby increase exercise tolerance [60]. Aerobic exercise has important advantages, particularly in increasing endothelial function and improving vasodilatory capacities [14]. It has been demonstrated that Tai Chi, a type of low-intensity aerobic exercise, could significantly enhance isokinetic quadriceps strength [11] and knee extensor muscle strength [52] in HF patients. Yeh et al. have found that, compared to conventional aerobic exercise, Tai Chi showed more significant improvements in 6MWD in HFpEF patients [14]. The potential mechanism of Tai Chi or other TCEs might be strengthening skeletal muscle, stretching, and enhancing neuroregulation.

In HFrEF patients, the peak VO2, an indicator of cardiopulmonary exercise capacity, is the strongest prognostic predictor of survival. For every 1 mL/min/kg lower peak VO2, there is a 16% greater risk of all-cause mortality [61]. Although several meta-analyses reported that ExCR [62], Tai Chi, and Qigong practices [58] improved peak VO2, these findings were inconsistent in the present meta-analysis and other previous studies [52,59,63]. Reduced oxygen consumption during TCE might be related to the lower heart rate since oxygen consumption is partly based on heart rate responses [14]. This divergence could also partially be elucidated by the substantial heterogeneity of study participants, including age, gender, disease history, etiology, myocardium damage, and cardiorespiratory fitness.

The results for BNP/NT-pro BNP and LVEF in this study are comparable to those of Ren’s [63], Pan’s [59], and Taylor-Piliae’s [57] studies, illustrating a significant reduction in circulatory BNP/NT-proBNP levels and improvements in LVEF. TCE, such as Tai Chi, Wuqinxi, Baduanjin, and Liuzijue, could activate muscles, dredge meridians, and collaterals, and regulate the circulation of Qi and blood [64]. In addition, TCE has been substantiated to inhibit sympathetic activity, activate the parasympathetic nerve, and increase coronary collateral circulation, cardiac output, and cardiac stroke volume [63]. Baduanjin could decrease inflammatory factors and reverse ventricular remodeling in patients with CHF [25]. Therefore, TCE has sound regulative effects on BNP/NT-pro BNP and LVEF.

QoL is evaluated by a variety of instruments, with MLHFQ being one of the most commonly used. The result of MLHFQ in this study is somewhat consistent with published systematic reviews that examined the effects of Tai Chi [57,59,63] and Qigong [58] on patients with HF, showing an improvement in QoL. It has been reported that ExCR could counteract the adverse effects of HF on psychological health, further stabilize physical fitness, and increase QoL in patients with CHF [62]. TCEs are characterized by a steady rhythm, inherent slowness and stillness, gentle movement, and breathing techniques in a harmonious manner [10,25,39], resulting in less cardiopulmonary stress and total body relaxation. Subsequently, TCE could alleviate HF symptoms, thus improving QoL.

Like other exercise programs, longer TCE duration has a better effect on exercise tolerance and cardiac function, which confirms the previous hypothesis [16,17,18]. In long-term CR for HF, lower intensity form exercise may produce less ventricular wall stress and lower risk of malignant arrhythmias [65]. In addition, long-term low-intensity activity appears to be as effective as that of moderate- to high-intensity in promoting muscle strength, autonomic nervous function, aerobic capacity, and QoL [66].

Breathing exercises could reduce the risk factors of cardiovascular disease through relaxation techniques and breathing modification [63]. Six sounds (Xu, He, Hue, Si, Tap, and Xi) are produced with different body movements when practicing Liuzijue. This type of breathing could slow the expiratory flow rate, especially facilitating gas exchange in patients with chronic obstructive pulmonary diseases [67]. Reduced breathing rate during Tai Chi indicated that the training effects differed from that of aerobic exercise [14]. A recent study found that diaphragmatic (abdominal) breathing, a fundamental procedure during meditation in Tai Chi, could improve dyspnea and daily physical activity, and further maintain or improve the cardiorespiratory fitness of patients with HF [68].

Distinguished from conventional aerobic exercises, such as the treadmill or bicycle workouts, TCE incorporates both physical and meditative elements [37], which are diverse and complex means of mind-body intervention [69]. TCE emphasizes the combination of physical postures, mind, and breathing techniques in a harmonious manner [70]. By practicing TCE, trainers can increase their self-awareness and self-efficacy, improve concentration, and relax [71]. These characteristics might impact the effect of cardiac rehabilitation in patients with CHF. Moreover, compared to aerobic exercise, TCE’s relatively lower intensity and meditative qualities may help accessibility and promote longer-term exercise adherence [14,72,73]. According to the study by Belardinelli et al., the LVEF of exercise-trained patients with CHF increased after 5 years of training duration [74]. It may be one of the reasons why the increases in LVEF presented in patients with CHF who received TCE rather than those who received conventional aerobic exercise. A previous study has explored that both aerobic and resistance training and TCE could improve the cardiopulmonary aerobic exercise capacity (e.g., peak VO2) of patients with stable coronary heart disease at different degrees [75]. Subgroup analysis of our study’s exercise program provides evidence that combining TCE with traditional aerobic exercise improves the exercise tolerance of patients with CHF more efficiently than aerobic exercise alone. Since only two studies performed combined exercise programs, further studies are still required.

This meta-analysis has several limitations. Except for two studies [14,40], blinded assessors were not recruited to evaluate echocardiographic results in most of the selected studies, which could increase interpretation bias. Adherence might be an essential factor and might be culture-dependent since the predominance of studies were performed in Asian countries. Limited by the number of studies, we could not analyze the effect of TCE on different phenotypes of CHF. Although the robustness of the results has been confirmed in a sensitivity analysis, some studies might have a bigger impact, especially in the subgroup analysis of exercise programs; therefore, more TCRs are required. Some factors may influence the results of this study, including the heterogeneity of TCE models in selected studies, short intervention and follow-up time, and the lack of data on recurrence rate, mortality rate, and cardiovascular endpoint event rate in follow-up data of patients in the TCE group. Therefore, this study failed to systematically evaluate TCE’s long-term clinical effects. Given that the sample size of some forms of exercise (e.g., Liuzijue, combined exercise program) is limited, there is a dire need for further studies to expand our knowledge.

## 5. Conclusions

As a safe and effective form of exercise, TCE significantly improves exercise capacity, cardiac function, and QoL in patients with CHF. With prolonged exercise duration, 6MWD was significantly elongated. TCE or TCE combined with conventional aerobic exercise improves the exercise tolerance of patients with CHF more efficiently than aerobic exercise alone. Thus, TCE might be an optimal and promising pattern of exercise-based cardiac rehabilitation for patients with CHF.

## Figures and Tables

**Figure 1 jcm-12-02150-f001:**
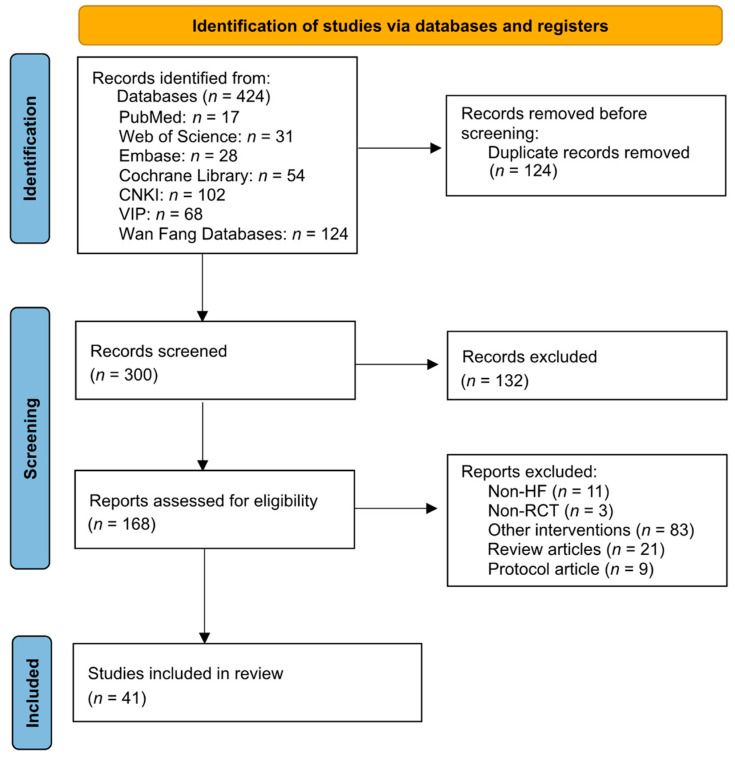
PRISMA flow chart of studies through the review. CNKI: the China National Knowledge Infrastructure; HF: heart failure; RCT: randomized controlled trial; PRISMA: Preferred Reporting Items for Systematic Reviews and Meta-Analyses.

**Figure 2 jcm-12-02150-f002:**
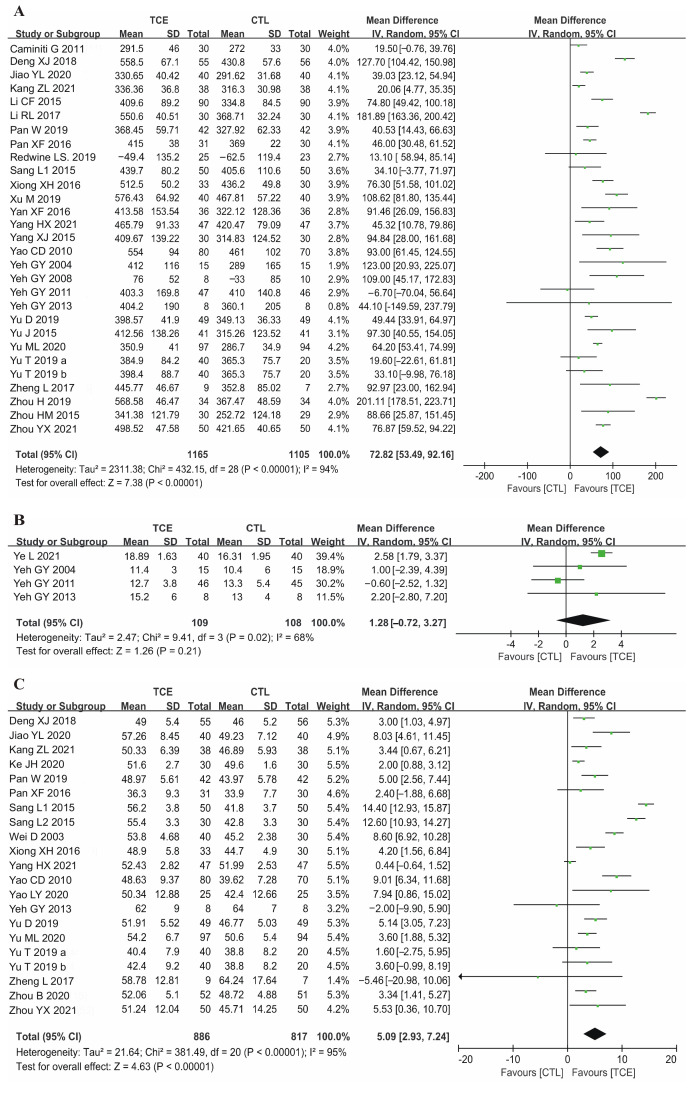
Forest plot for the effect of TCE on patients with CHF: (**A**) 6-min walk distance (6MWD) [11,12,13,14,16,17,18,19,26,27,29,32,33,34,36,37,38,39,41,43,44,45,46,47,48,49,50,51,52], (**B**) peak oxygen consumption (peak VO2) [14,31,37,39], (**C**) left ventricular ejection fraction (LVEF) [12,14,15,16,19,25,27,28,29,30,32,44,45,46,47,48,49,50,51,53]. TCE—traditional Chinese exercises; CHF—chronic heart failure; CTL—control; SD—standard deviation; CI—confidence interval. The green square represents the point estimate of each study. The size of the square represents the “weight” of the study. The black rhombus represents the pooled effect estimates and CI from all studies included in the meta-analysis.

**Figure 3 jcm-12-02150-f003:**
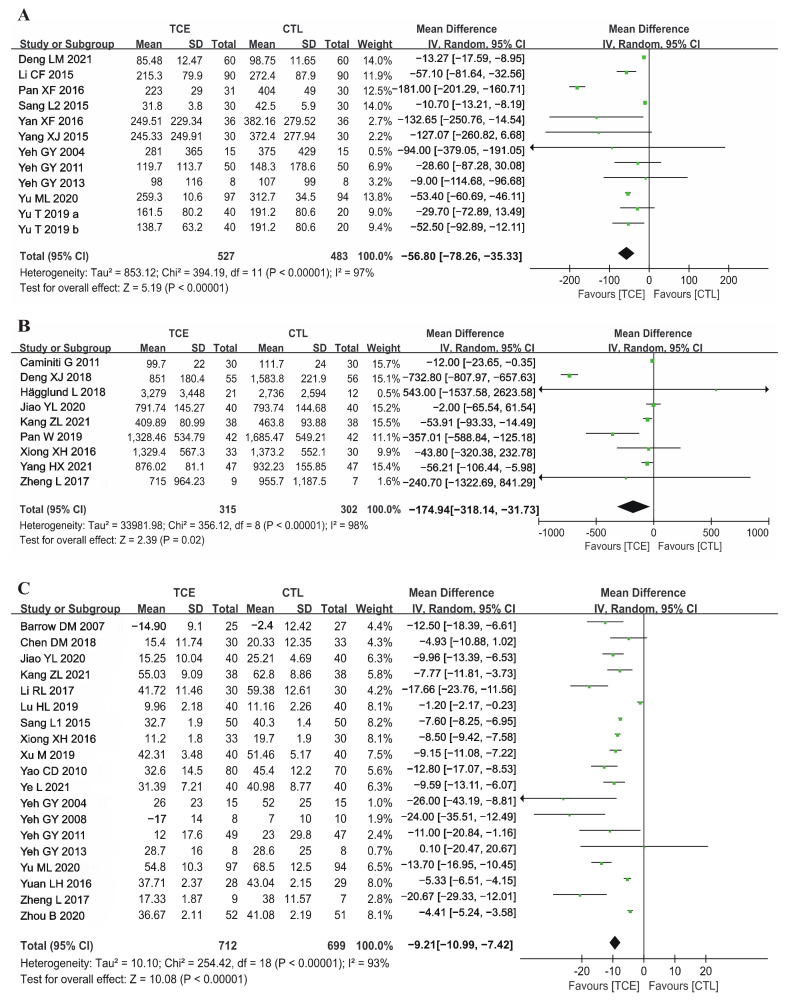
Forest plot of meta-analysis in other effects of TCE on patients with CHF: (**A**) B-type natriuretic peptide (BNP) [14,17,18,34,37,39,44,45,48,53,55], (**B**) N-terminal pro-B natriuretic peptide (NT-proBNP) [11,12,16,19,27,41,46,47,54], (**C**) Minnesota Living with Heart Failure Questionnaire (MLHFQ) [10,12,14,15,16,19,24,26,29,31,34,37,38,39,42,43,46,49,56]. TCE—traditional Chinese exercises; CHF—chronic heart failure; CTL—control; SD—standard deviation; CI—confidence interval. The green square represents the point estimate of each study. The size of the square represents the “weight” of the study. The black rhombus represents the pooled effect estimates and CI from all studies included in the meta-analysis.

**Figure 4 jcm-12-02150-f004:**
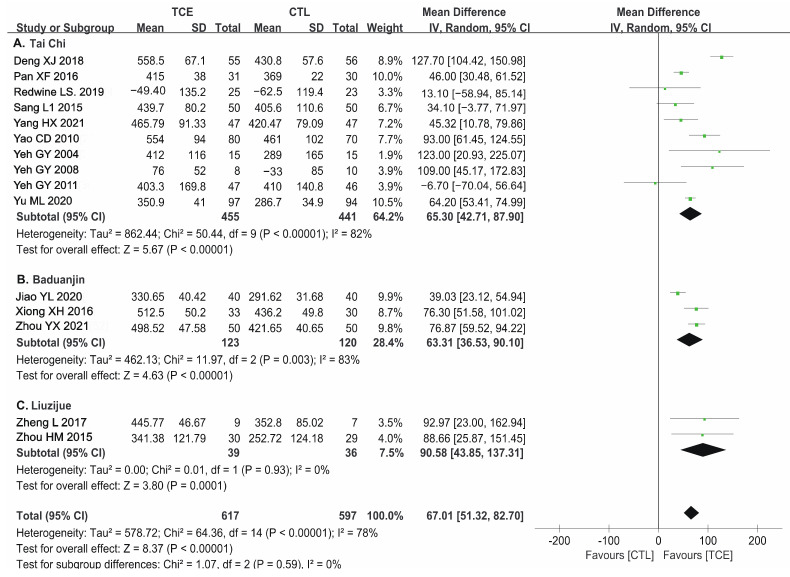
Forest plot for the subgroup analysis of TCE types on 6MWD: (**A**) Tai Chi [29,34,37,38,39,41,47,48,49,54]; (**B**) Bauduanjin [12,19,50]; (**C**) Liuzijue [13,16]. TCE—traditional Chinese exercises; 6MWD—6-min walk distance; CTL—control; SD—standard deviation; CI—confidence interval. The green square represents the point estimate of each study. The size of the square represents the “weight” of the study. The black rhombus represents the pooled effect estimates and CI from all studies included in the meta-analysis.

**Figure 5 jcm-12-02150-f005:**
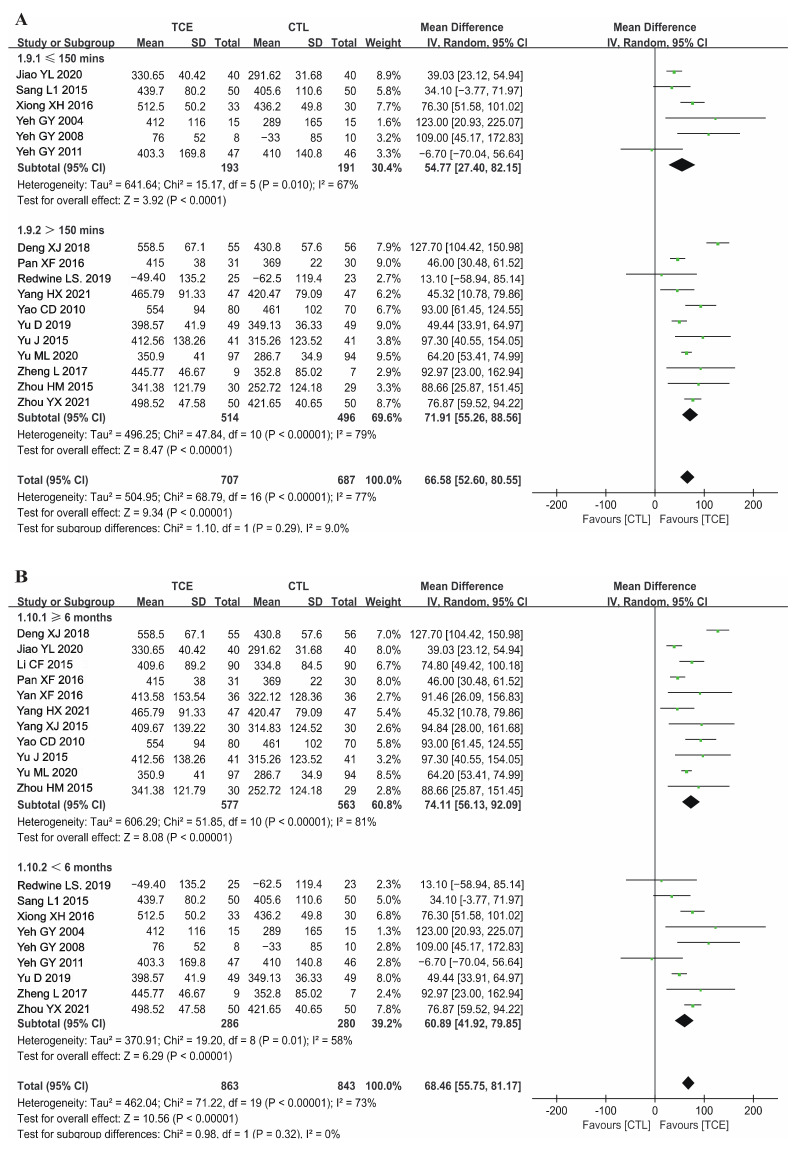
Forest plot for the subgroup analysis of TCE weekly duration and program duration on 6MWD: (**A**) weekly duration [12,13,17,18,29,33,34,47,48,54,55], (**B**) program duration [16,19,32,37,38,39,41,49,50]. TCE—traditional Chinese exercises; 6MWD—6-min walk distance; CTL—control; SD—standard deviation; CI—confidence interval. The green square represents the point estimate of each study. The size of the square represents the “weight” of the study. The black rhombus represents the pooled effect estimates and CI from all studies included in the meta-analysis.

**Figure 6 jcm-12-02150-f006:**
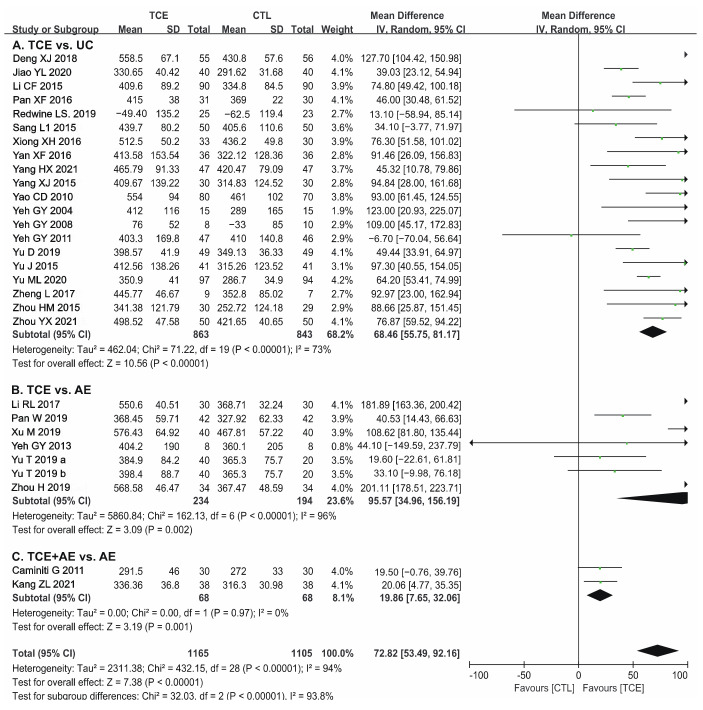
Forest plot for the subgroup analysis of exercise program on patients with CHF on 6MWD: (**A**) TCE compared with usual care [12,13,16,17,18,19,29,32,33,34,37,38,39,41,47,48,49,50,54,55], (**B**) TCE compared with conventional aerobic exercise [14,26,27,36,43,44,45], (**C**) TCE and conventional aerobic exercise compared with conventional aerobic exercise [11,46]. CHF—chronic heart failure; 6MWD—6-min walk distance; TCE—traditional Chinese exercises; CTL—control; UC—usual care; AE—aerobic exercise; SD—standard deviation; CI—confidence interval. The green square represents the point estimate of each study. The size of the square represents the “weight” of the study. The black rhombus represents the pooled effect estimates and CI from all studies included in the meta-analysis.

## Data Availability

The protocol was registered in the International Platform of Registered Systematic Review and Meta-analysis Protocols (INPLASY): Invoice Number: 202190027.

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
