# Peer review of "Effect of Traditional Chinese Exercises on Patients with Chronic Heart Failure (TCE-HF): A Systematic Review and Meta-Analysis"

_jcm, 2023, doi:10.3390/jcm12062150_

Round 1
Reviewer 1 Report
I read the article with interest. With the high prevalence of heart failure and the limitation of the quality of life that it entails, rehabilitation plays an important role. A systematic review and meta-analysis bring a relevant contribution to knowledge on the subject. In this case, the differential intervention is rehabilitation usingtraditional Chinese exercises versus conventional treatment.
The article is well organized and easy to read, being a meta-analysis respecting the Cochrane Handbook and Preferred Reporting Items for Systematic Reviews and Meta-Analyses (PRISMA)
Of the 424 studies initially identified, 41 were included. In geographical terms, there is a predominance of studies from Asian countries.
In addition to the impact of traditional Chinese exercises in general, measured in 6-minute walk distance (6MWD), peak oxygen consumption (peak VO2), left ventricular ejection fraction (LVEF), B-type natriuretic peptide (BNP), N-terminal pro-B natriuretic peptide (NT-proBNP), Minnesota Living with Heart Failure Questionnaire (MLHFQ) several subgroups were analyzed; by type of exercise (Tai Chi, Baduanjin, Liuzijue) and by duration
Some corrections are needed
Line 14 TCE is not spelled out
Page 6, has too many graphics. I suggest reorganizing with few tables per figure
Line 229 repeated heading (215)
Line 333 and following
I suggest including a note on the predominance of studies in Asian countries. Adherence will be an essential factor and may be culture dependent
Conclusions (line 342 and following)
Information regarding duration associated with positive results could be included
Author Response
We thank the Editors and the Reviewers for their constructive comments that have assisted us in improving this manuscript. The point-by-point response to the reviewer’s comments is submitted as attachment.

Reviewer 2 Report
First I would like to commend the authors for this work. It brings a very interesting topic in heart failure management. However I would like to address some issues to improve the work. I'll divide in minor and major issues sections.
Minor issues:
1- Abstract should follow the journal instructions. Instead of "introduction", the authors should replace for "background"
instructions for authors: "The abstract should be a single paragraph and should follow the style of structured abstracts, but without headings: 1) Background: Place the question addressed in a broad context and highlight the purpose of the study."
2- There is a duplicate section "3.5.2. TCE duration" on page 8. Same section is present on page 7. I believe this was not intentional.
3- there is a difference in font size in the risk of bias section (page 4), please correct.
4 - One of the risk of bias graphs is too small, please change to provide a better view.
Major issues:
Some manuscript's points called my attention, and I would like to address them.
1 - In the method's sections the authors states the sensitive analysis would be performed to elucidate the heterogeneity between studies. However, I did not find any section or description about the results of this analysis. I believe that they may play an important role on results and consequently on discussion.
1.1) For instance, the authors choose to perform together the analysis of left ventricular ejection fraction of both preserved and reduced ejection fraction heart failure patients. Since these are two different phenotypes of HF, with different pathophysiological mechanisms, I recommend a subgroup analysis to see if there is differences in response to the intervention.
1.2) Another issue that called my attention is in the 3.5.2 section, figure 5. The comparison TCE vs AE. Some of the studies included present very impressive improvements in terms of exercise tolerance (6MWD). Moreover, the effect size of TCE vs aerobic exercise is higher in comparison to the control group (95.57 meters - TCE vs AE and 68.46 TCE vs CTL) is there any explanation for this? How the effect size is affected when sensitive analysis is performed? Additionally, the authors should discuss this performance of TCE over aerobic exercise, since this is a well established intervention for heart failure patients.
1.3) The authors claimed that "visual inspection of the funnel plot for 158 each outcome suggested no visible publication bias". But, there is no Funnel plot available either on the manuscript nor in the supplemental material. The graphic should be in some of these files. please provide this figure.
In conclusion, I believe that addressing these major issues is of high importance to this work.
Author Response

(The authors gave the same response as above.)

Round 2
Reviewer 2 Report
Thank you for the corrections and congratulations on your work.